# Therapeutic Treatment of 2A Grade Burns with Decellularized Bovine Peritoneum as a Xenograft: Multicenter Randomized Clinical Trial

**DOI:** 10.3390/medicina58060819

**Published:** 2022-06-17

**Authors:** Berik Tuleubayev, Vyacheslav Ogay, Baurzhan Anapiya, Assylbek Zhylkibayev, Dina Saginova, Amina Koshanova, Yerkin-Dauir Kurmangaliyev, Kanat Tezekbayev, Nurzhan Bikonurov, Kabylbek Abugaliyev

**Affiliations:** 1Surgical Diseases Department, Karaganda Medical University, 40 Gogol str., Karaganda 100000, Kazakhstan; berik-karaganda@rambler.ru (B.T.); koshanova87@mail.ru (A.K.); 2Stem Cells Laboratory, National Center for Biotechnology, 13/5 Korgalzhyn Ave., Nur-Sultan 01000, Kazakhstan; ogay@biocenter.kz; 3“X-Matrix” LLP, 13/5 Korgalzhyn Ave., Nur-Sultan 01000, Kazakhstan; abkar@mail.ru; 4Department of Multidisciplinary Surgery, National Research Oncology Centre, 3 Kerei-Zhanibek Khandar str., Nur-Sultan 01000, Kazakhstan; 5Department of Optometry and Vision Science, The University of Alabama at Birmingham, Birmingham, AL 35294, USA; askokshe@gmail.com; 6Laboratory of Applied Genetics, National Center for Biotechnology, Nur-Sultan 01000, Kazakhstan; 7National Scientific Center of Traumatology and Orthopedics Named after Academician N.D. Batpenov, 15a Abylay khan Ave., Nur-Sultan 01000, Kazakhstan; sa_dina@mail.ru; 8Multidisciplinary Hospital named after Professor H.Zh. Makazhanov, 5/3 Mukanov str., Karaganda 100000, Kazakhstan; dakekz@mail.ru; 9Department of Traumatology and Orthopedics, Kazakh National Medical University, 220 Papanina str., Almaty 050000, Kazakhstan; tekanat@mail.ru; 10Burn Department, Municipal Clinical Hospital №4, Almaty 050000, Kazakhstan; bikonurov@inbox.ru

**Keywords:** decellularized bovine peritoneum, xenograft, burns, wound dressing, regeneration

## Abstract

*Background and Objectives:* Homogeneous and xenogenic bioengineering structures are actively used as wound coatings in treatment of burns and have already shown their effectiveness. Nevertheless, the disadvantage of such dressings is their high cost. This issue is particularly challenging for developing countries in which the incidence of burns is the highest one. With such needs taken into account, the research team developed and clinically tested a new wound coating based on decellularized bovine peritoneum (DBP). *Materials and Methods:* A multicenter randomized clinical trial was conducted to evaluate DBP. The following variables were considered in the research study: the number of inpatient days, the number of dressing changes, the level of pain experienced during dressing changes, and the condition of wounds at the time of the follow-up examination. *Results:* The research involved 68 participants. It was found that the patients who were treated with a DBP experienced less pain with less changes of dressings. However, the number of inpatient days and wound healing failed to demonstrate statistically significant difference compared to the control group. *Conclusions:* In the given research, DBP showed efficacy in improving patients’ quality of life by reducing pain and the number of dressings’ changes. However, when comparing this research study with the studies of other animal-derived wound coverings, there were a number of differences and limitations in the parameters. Thus, the results requires further study for a greater comparability of data. Given the above, we expect that DBP will become an inexpensive and effective treatment for burns in developing countries.

## 1. Introduction

According to the World Health Organization, burns cause 180,000 deaths per year [1]. Even non-fatal cases of burns pose a significant health hazard to patients in the form of developing infections and contractures [2].

Partial thickness burns were treated conservatively by wound debridement followed by local use of drugs such as an ointment or dressing sheets [3]. However, this method of treatment can be very painful for a patient and can cause stress disorders [4]. A modern method of conservative treatment of II-III-degree burns is the use of biological wound dressings [5]. Both homogeneous and xenogeneic biological dressings are currently used to treat partial thickness burns such as Alloderm^®^, Transcyte^®^, Integra^®^, Apligraf^®^, Human amnion, Oasis^®^, and Biobrane^®^ [5,6,7,8,9]. Clinical studies have shown that these wound dressings can speed up the healing process, prevent infections from entering a wound, and reduce the level of pain experienced by a patient during dressing change [5]. However, one of the disadvantages of these commercially available wound dressings is their high cost, which limits their usage in developing countries, including Kazakhstan. For example, the 2 × 2 inches and 4 × 5 inches Integra^®^ wound dressing would cost USD 1400 and USD 2205, respectively [10].

In this regard, we have developed a technology for obtaining a relatively inexpensive wound dressing from the parietal peritoneum of a bovine for applications to treat burn wounds [11]. The parietal peritoneum represents a connective tissue layer covering the abdominal wall and diaphragm. The connective tissue layer of the parietal peritoneum mainly consists of collagen and elastin fibers, which provides it with high strength and elasticity [12]. To obtain a xenogenic wound dressing, we found optimal technological conditions for the decellularization of the parietal peritoneum of a bovine and radiation sterilization. Our preclinical studies have shown that the xenogenic wound dressing obtained from the parietal peritoneum is safe. It also accelerates the regeneration of extensive full-layer wounds in experimental rats. Thus, taking our pre-clinical data into account, a pilot production of a xenogenic wound dressing called X-Graft^®^ was launched according to the ISO 13485 international standard. Official permission was obtained from the Ministry of Health of the Republic of Kazakhstan to conduct a randomized controlled clinical trial involving the patients with extensive burn wounds [13].

The main aim of the clinical trial was to evaluate number of inpatient days, pain range, and either the presence or absence of complete wound healing in the patients with IIA grade burn wounds treated with X-Graft^®^ wound dressings.

## 2. Materials and Methods

The decellularized bovine parietal peritoneum as a wound dressing was provided by X-Matrix LLP (Kazakhstan).

### 2.1. Preparation of Decellularized Parietal Bovine Peritoneum

The Bovine parietal peritoneum (BPP) was processed according to the detergent enzymatic method. Briefly, BPPs were washed in phosphate buffered saline (PBS) containing 1% povidone-iodine (PI). Afterward, BPPs were washed twice in distilled water (containing 1% PI) and twice in Milli-Q water containing 1% penicillin/streptomycin/amphotericin B. In order to remove cell debris, BPPs were treated with 4% sodium deoxycholate for 4 h at room temperature and washed twice in distilled water. This process was repeated twice. To completely remove cell DNA, BPPs were incubated with DNase I (2000 KU) in 1 M NaCl in a shaker–incubator for 3 h at 37 °C and then washed three times in distilled water. The decellularized BPPs were packed in the blister packs and treated by β-radiation at sterilizing dose of a 16 kGy.

### 2.2. Histological Analysis

The BPP samples were fixed in a 10% solution of neutral formalin for 24 h at room temperature. The samples were sequentially dehydrated in 70%, 95%, 95%, 100%, and 100% ethyl alcohol after washing in phosphate buffer and then they were immersed in xylene. After that, the samples were infiltrated with paraffin and enclosed in paraffin blocks. Histological sections with 5 μm thickness were cut on a microtome (Leica, Germany) and applied to a glass slide. The slides with paraffin sections were treated twice with xylene to remove paraffin. Further on, the samples were rehydrated according to the following scheme: 100%, 100%, 95%, 95%, and 70% ethyl alcohol and distilled water. Thereafter, the samples were stained with hematoxylin, washed with water, stained with eosin, and sequentially dehydrated with ethyl alcohol 95%, 95%, 100%, and 100%. The histological medium was applied to the slides, and each slide was covered with a cover slip after processing with xylene. The stained samples were analyzed with the Axio Scope A1 light microscope (Carl Zeiss, Jena, Germany).

### 2.3. Participants

The research team has conducted an open, controlled, and multicenter randomized clinical trial at the premises of the burn centers of the City Clinical Hospital No. 4 in Almaty and the Professor Kh. Zh. Makazhanov Multidisciplinary Hospital in Karaganda from September 2019 to November 2021. The research program was approved by the Meeting of the Central Ethical Commission of the Republic of Kazakhstan on 25 January 2019.The research was conducted in accordance with the Declaration of Helsinki and amendments to it, registered in the Committee for Control of Medical and Pharmaceutical Activities of the Ministry of Health of the Republic of Kazakhstan as of 06.03.2019 under the identification number No. KZ90VMX00000189. 

The research participants were recruited from the patients of the burn centers. Both women and men were included in the research study. The inclusion criteria were as follows: age ≥18 and ≤60 years, possessing dermatological wounds caused by superficial partial thickness burns determined by observation of burn surgeon affecting from 5% to 30% of the total surface area of the body, the lack of previous treatment for current burns, and the absence of other significant diseases that may affect the volunteer’s participation in the research study (coronary heart disease, peripheral vascular disease, cancer, diabetes, etc.). 

The exclusion criteria included hypersensitivity to the materials used in the research or to related drugs, severe allergic reactions in the anamnesis, drug dependence, the use of drugs that can affect the process of wound healing (such as steroids), and pregnancy.

Moreover, patients with burns on the face, neck, genitals, perineum, armpits, groin, and buttocks were not included due to the difficulties in applying decellularized peritoneum on these areas. 

The informative conversation was conducted with patients meeting the inclusion criteria for the given research study, with informed consent obtained to participate in the research study. After this procedure, the patients were randomized. The test or research group received treatment with a decellularized bovine peritoneum, and the control group’s patients received treatment using gauze dressings impregnated with Povidone-Iodine 10%^®^ (Medoptik LLP, Almaty, Republic of Kazakhstan). The diagram showing the design of the research is shown below in Figure 1.

### 2.4. Interventions

The patients of the research group were treated by applying a decellularized bovine peritoneum (DBP) to a burn wound. In the control group, gauze dressings impregnated with 10% Povidone-Iodine were used. This material was selected due to the fact that all burn centers in Kazakhstan use Povidone-Iodine 10% as a standard means to treat burn wounds [14].

After physical examination, anthropometric parameters and vitality assessment, and the evaluation of total burn surface area by using the Lund–Browder chart, the patients were assigned to one of the treatment groups (research or control groups) by randomization.

The patients were randomized by a computer by generating and assigning each patient an individual number. Each patient with an individual even number was assigned to the control group, patients with odd numbers joined the research group. 

After enrollment in the research study, the patients received standard treatments according to their placement. The patients in both groups underwent mechanical debridement of the wound from external contaminants, and superficial necrotic tissue and blisters were removed. After that, 2% chlorhexidine gluconate was used to treat the wounds [15].

Further on, the patients of the test group were applied with DBPs. DBPs were fixed by surface bandaging with a gauze roll bandage without suturing or stapling. During the entire period of treatment, the DBPs were only changed when there was a lack of adhesion or some accumulation of exudates under the dressing. DBPs were removed when patients developed an allergic reaction to the studied dressing material. 

To assess the state of DBP, a gauze bandage was removed every 48 h. The level of dressing’s adhesion, the presence or absence of accumulated exudate under the dressing, and the level of pain experienced by the patient have been evaluated. In the absence of an adequate level of adhesion or accumulated exudate under the dressing, fixing gauze bandages were reapplied. In cases of a satisfactory general condition of a patient (stable vital and laboratory parameters and complete adhesion of DBP), this patient was discharged from the hospital. A follow-up examination of a patient was carried out on the 14th day after discharge from the hospital. Graft loss or the product not integrating into the wound was assessed. In cases of a complete discharge of the xenograft, the presence or absence of residual wounds was also assessed. 

The gauze bandages with 10% Povidone-Iodine were applied in the control group. These gauze dressings were fixed by superficial bandaging with a gauze bandage. The gauze dressing was changed each time it was soaked (impregnated) with wound discharge or every 48 h in the absence of dressing’s wetting. The bandage was no longer applied in case of a complete epithelialization of the wound.

The wound under a gauze dressing with 10% Povidone-Iodine was assessed every 48 h. The levels of dressing adhesion, wetting of the dressing, and the appearance or absence of granulation and epithelialization spots were assessed. A follow-up examination of a patient was carried out on the 14th day after discharge from the hospital. The presence or absence of residual wounds was assessed.

The patients were assessed by clinical and research parameters every 48 h up to the moment of discharge from the hospital. The patients are discharged from the hospital with stable clinical and laboratory indicators, the complete adhesion of wound coatings, and the satisfactory wellbeing of the patient. A follow-up examination of a patient was carried out on the 14th day after discharge, and all patients were monitored by the same research team. 

### 2.5. Variables

The primary variables of treatment outcomes were the presence of complete re-epithelization observed by the clinical consultant on the 14th day after the patient’s discharge from the hospital. Moreover, the presence of DBP sheets adhered to the wound bed in the research group was regarded as the absence of complete re-epithelialization. 

The secondary variables of treatment outcomes were as follows: the degree of pain during dressing application and changes, the number of dressing changes, and the number of days spent by patients in the hospital.

The level of pain was assessed across 5 various options on an analogue scale, which was approved as part of the clinical trial program by the Central Commission on Ethics under the Ministry of Health of the Republic of Kazakhstan dated 25 January 2019 (Protocol No. 1).

After applying dressings and during the change of dressings in the control group as well as during the follow-up examination in the research group, each patient was asked to describe the pain experienced according to one of the criteria and to fill out a table. Afterward, the results were ranked from “No pain”—0 points—to “Intolerable pain”—4 points (Table 1).

The dressings were changed every 48 h in the control group, and the change procedure consisted in the complete removal of gauze dressings, exudate, tissue plaque if any, and repeated applications of gauze dressings with 10% Povidone-Iodine. In cases of abundant wetting of the dressings or accumulations of tissue effusion, the bandages were replaced more often than 48 h. The replacement of one or more DBP sheet was considered as a dressing change. In cases of adhesion absence, with the presence of accumulated exudate under the dressing, the DBP sheets were replaced more often or less than 48 h.

The number of inpatient days was calculated by the number of calendar days spent by a patient in the burn center from the moment of admission to the moment of discharge.

### 2.6. Statistical Analysis

The statistical analysis was carried out by using Microsoft Excel^®^, Microsoft, New York, NY, USA, IBM SPSS Statistics^®^ V 26, Armonk, NY, USA.

Sample size was calculated with a two-sided t-test performed with a 0.05 type 1 error, power of 0.80–0.90, and effect size 0.8.

For descriptive statistics, average values with standard deviation were used. For the ease of interpretation and use, all data are rounded to 0.00. The statistically significant value was considered at *p* value > 0.05.

The quantitative variables were initially analyzed using the Kolmogorov–Smirnov test for normal distribution. 

The statistical analysis of quantitative data with normal distribution was carried out using the Student’s *t*-test criterion for independent samples. 

For quantitative data with an abnormal distribution, the nonparametric Mann–Whitney U criterion was used. To analyze the statistical difference in the qualitative data in the studied samples, the nonparametric criterion of Pearson’s chi-squared test was used.

## 3. Results

### 3.1. General Data and Comparability of Groups

In total, the research study involved 68 patients: 31 in the test and 37 in the control group. Of these, 47 are men (27 in the control group and 20 in the test group) and 21 were women (10 in the control group and 11 in the test group). The differences in the distribution of men and women into groups did not show a statistically significant difference. 

According to the results of the Kolmogorov–Smirnov test, it was revealed that sampling by age had a normal distribution. The indicators of the area, the number of inpatient days, the number of dressing changes, and pain ranges during the application and changes of dressings did not have a normal distribution. 

The degree of burn of patients in both groups was the same as it was one of the main criteria for inclusion in the test group of the given research. 

The statistical analysis of the age and the area of burns did not show a significant difference. The results of the above tests allowed us to assert that the studied groups are comparable to each other. 

No allergic reaction cases to DBP or Povidone-Iodine 10% were detected in both groups.

### 3.2. Number of Inpatient Days 

The amount of time spent by patients in the burn center was calculated by the number of calendar days from the moment of hospitalization in the burn center. The analysis showed that the use of a decellularized bovine peritoneum did not reduce the inpatient time spent by the patient in the burn center (*p* ≈ 0.50).

The statistical analysis of the number of days spent by patients in the conditions of the burn center did not have a statistically significant difference in both groups (*p* ≈ 0.51). The decellularized bovine peritoneum does not reduce inpatient times.

### 3.3. The Level of Pain Experienced

The level of pain experienced in each group was measured at the time of the application and during each subsequent doctor’s visit. The level of pain experienced is pre-coded in accordance with the analogue scale shown in Table 1. Statistical analysis was carried out between each visit in both groups, up to visit 5. This is caused by the duration of the patients’ stay at the burn center, which caused a different number of visits and dressing changes and a decrease in statistical data after Visit 5 (Table 1). 

The result of the statistical analysis showed that patients in the test group experienced a less intense feeling of pain compared to the control group in each visit (*p* < 0.05).

### 3.4. Frequency of Dressing Changes

The study showed that the patients in the test group were much less likely to change dressings (*p* < 0.05). All data are presented in Table 2.

### 3.5. Healing Analysis

At the time of the follow-up examination, one could observe complete re-epithelization in 47 out of 68 patients. Of these, 23 were in the test group and 24 were in the control group. The remaining subjects at the time of the follow-up examination had small residual wounds or fragments of applied dressings, which were regarded by the research team as a lack of complete re-epithelialization. 

Statistical analysis performed for the healing process also did not show a significant difference between the control and test groups (Table 2) (Figure 2).

## 4. Discussion

The inpatient duration in the test group at the burn center averaged 10.45 and 9.92 days in the control group, which did not have a significant difference in the statistical analysis (*p* ≈ 0.50). However, it is worth taking into account that the xenobiological dressings were used in the territory of Kazakhstan for the first time. In practical terms, it caused a certain level of caution from the doctors involved in the treatment of patients. Thus, the inpatient treatment of the patients of the test group was prolonged for an additional period at the burn center for more detailed observation.

This parameter is important for us from an economic point of view since the financial compensation of each treated case has a fixed cost irrespective of the duration of the patient’s stay in the hospital and the amount of treatment received [16]. Reducing the inpatient time at the burn center will save money and provide an opportunity to invest in future research.

The analysis of the wounds’ epithelialization did not show satisfactory results. Statistical analysis showed no difference in complete wound epithelialization between the test and control groups at the time of follow-up examination in 14 days after discharge (*p* ≈ 0.69). Combined with the average inpatient period in the conditions of the burn center, it comprises 24 days. These results are worse if compared with other xerographs such as Nile tilapia skin, which allows an average of 9.77 ± 0.83 days and 10.56 ± 1.13 days [17] for re-epithelization, and porcine xerograph with re-epithelization at 13.22 ± 2.1 days [18]. A dermal substitute composed of a cross-linked bovine tendon collagen-based dermal matrix linked with glycosaminoglycans (Integra™) is also used for burn treatment. The duration of Integra ranges from 10–14 days to 8 weeks, with an average result of 30 days used in the treatment of deep partial thickness and full thickness burns. [19]. Biobrane^®^ biosynthetic material used in partial thickness burns in children allows an average of 9 ± 5 days for wound healing [20].

However, it is worth considering the fact that when using Nile tilapia’s skin, complete healing was considered at ≥95% of wound re-epithelization, [17,21] while in our study, only 100% re-epithelialization was considered as the criterion for healing. Even the presence of residual adhesive decellularized bovine peritoneum or gauze dressing with full functional recovery of the patient coupled with his or her ability to return to work and satisfactory wellbeing were recognized by the research group as the presence of a residual wound. 

There were some significant differences in the results in the number of dressing changes and the level of pain experienced by patients during dressing changes. 

This parameter is crucially important as any long-term pain caused by burns can cause a depressive state in the patient [4,22] and can undermine the trust relationship between the doctor and the patient. Such circumstances have a negative impact on the treatment as a whole [23].

The wound coverings of biologically derived tissue, such as the skins of pigs, Nile tilapia, and cadaver have long been used as xenografts for the treatment of burns [6,24,25].

However, their implementations and applications within the territory of the Republic of Kazakhstan are problematic due to their high cost and the lack of funds for their purchase in the local budget [16]. Harvesting and transplantation of cadaveric skin falls under the jurisdiction of the law “On Approval of the Rules for the Removal, Preservation, Transplantation of Tissues and (or) Organs (Parts of Organs) from Person to Person, from Corpse to Person and from Animal to Person”, which creates significant organizational and legal difficulties [26].

As an alternative for such dressings, bovine peritoneum was chosen. Bovine is the most common animal in agriculture and is distributed throughout Kazakhstan [27]. This brings the market cost of DBP to about USD 15 per 10 × 10 cm sheet, which is cheaper than Integra^®^, which is also made from bovine collagen, and Biobrane^®^ biosynthetic dressing. [10,28]. In the parietal peritoneum of mammals, the structure is similar to each other and has a basement membrane, which is a fibroelastic tissue containing glycosylated proteins, mast cells, macrophages, and lymphocytes [12]. 

In preclinical trials, DBP has shown that the collagen fibers obtained as a result of decellularization are similar in structure to human ones [29] (Figure 3).

In addition, promising results have been obtained with the use of DBP on rats. Wound healing in animals treated with DBP is much better than with gauze bandages impregnated with 0.05% chlorhexidine bigluconate [30]. Interestingly, DBPs have also demonstrated some good resistance to stretching [31].

The given research shows that the use of decellularized bovine peritoneum of cattle to treat burns reduces the intensity of pain and the number of dressing changes in patients. However, it is worth bearing in mind that 10% Povidone-Iodine is painful when applied. Thus, the decrease in pain in the test group may not be due to a direct effect of DBP on the healing process but to a decrease in the number of dressings. Moreover, parameters such as the number of inpatient days and the timing of re-epithelization require further and more detailed study.

This research had a number of limitations. The majority of patients live in regions located far away from burn centers and did not allow regular outpatient monitoring of the wounds. Taking these circumstances into account, the research group decided to perform a control examination of all patients on the 14th day after discharge. However, this approach of analysis does not allow recording earlier wound healing. Moreover, the COVID-19 pandemic and the movement restrictions applied in the regions due to the epidemiological situation limited the admission of patients from other regions, which significantly increased the duration of the present research study. A significant difference between the tested wound dressing and control dressing as well as the absence of the need for its regular change made it impossible to conduct a blind and double-blind study.

Furthermore, after applying DBPs to the wound, the tested wound adhered and dried out with the formation of a crust in 2–3 days after the application. This limited the movement of patients, and made the use of a decellularized bovine peritoneumon the face and at the site of physiological folds almost impossible. Therefore, the study involved patients with burn wounds on a stable surfaces like abdominal wall, thorax, hip, tibia, arm and forearm retreating from the joints. Such way of application did not limit patient’s movement.

## 5. Conclusions

In the given randomized controlled clinical trial, which included 68 patients, decellularized bovine peritoneum showed good adhesion to the wound bed, reducing pain, the number of dressing changes, and fluid loss. However, the analysis of the number of inpatient days and the level of epithelialization did not show statistically significant results, which requires further and thorough research. Given the above, it is expected that the decellularized bovine peritoneum will become inexpensive and an effective treatment for burns in developing countries, particularly those in Central Asia.

## Figures and Tables

**Figure 1 medicina-58-00819-f001:**
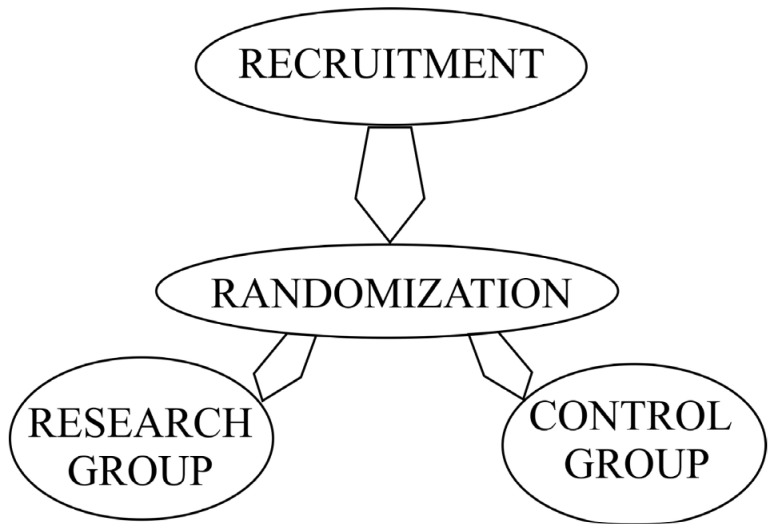
Research design.

**Figure 2 medicina-58-00819-f002:**
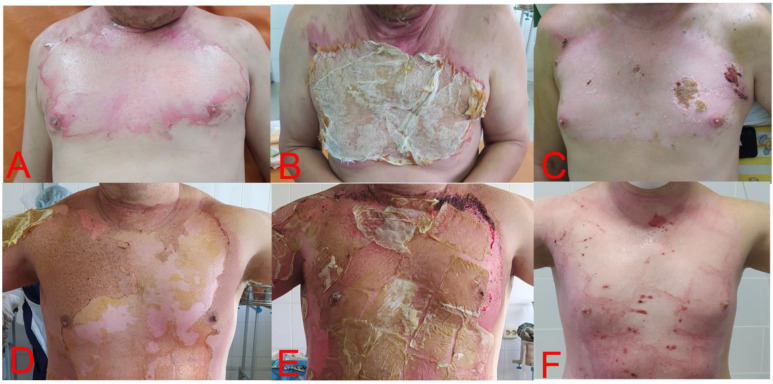
Participants of each group of study. (**A**) Patient of control group on the 1st day of trial, with 2A stage burns of anterior chest wall with boiled water, after debridement. (**B**) The same patient on the 3rd day of trial, and gauze bandages have adhered. (**C**) The same patient on the 14th day after discharge (20th day of trial). Re-epithelization of wound surface, with part of residual wound scab on the left side. (**D**) Test group patient on the 1st day of trial, with 2A stage burns of anterior chest wall with flame. (**E**) The same patient on the 3rd day of trial and total adherence of DBP, with drying. (**F**) The same patient on the 14th day after discharge (18th day of trial). Re-epithelization of wound surface. With small parts of residual wound scab on periphery of wound surface.

**Figure 3 medicina-58-00819-f003:**
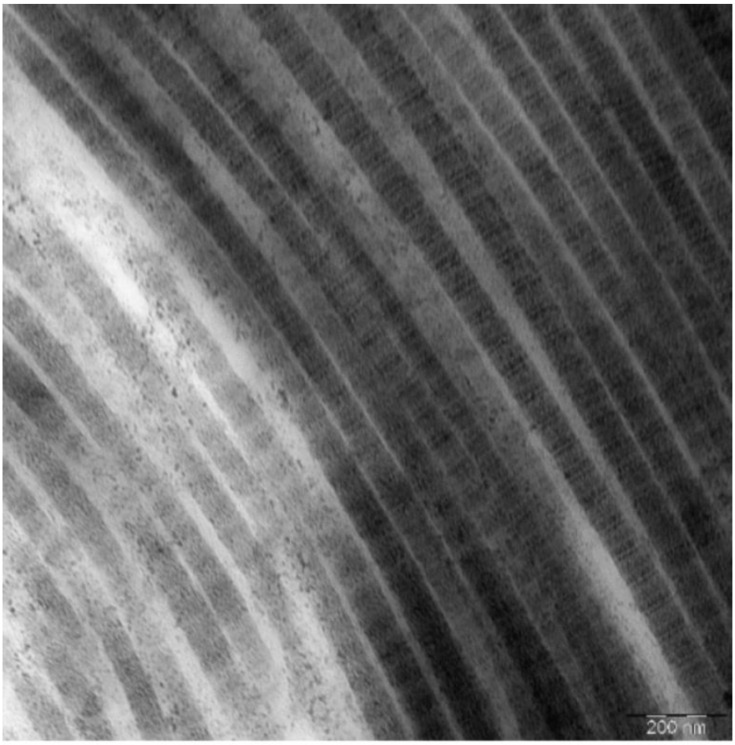
Electron micrograph of X-graft^®^ collagen fibers in longitudinal sections. Each fibril consists of regular alternating dark and light bands that are further divided by cross-striations. Scale bar is 200 nm.

**Table 1 medicina-58-00819-t001:** Analogue scale to range pain levels with schedule of patients’ examination.

Patient	Pain Character	Range	Visit 1 Date	Visit 2 Date	Visit 3 Date
XG 01.101	No pain	0		+	+
	Slight pain	1			
	Moderate pain	2	+		
	Severe pain	3			
	Intolerable pain	4			
XG 02.201	No pain	0			+
	Slight pain	1			
	Moderate pain	2		+	
	Severe pain	3	+		
	Intolerable pain	4			

XG01.101—patients individual code. 01—burn center code (01—Professor Kh. Zh. Makazhanov Multidisciplinary Hospital. Karganda; 02—City Clinical Hospital No. 4. Almaty). 1—group code. 1—test group, 2—control group. 01—patient’s sequential number. + pain level mark experienced during the visit.

**Table 2 medicina-58-00819-t002:** Clinical Research Statistics.

Parameter		Decellularized Bovine Peritoneum		Povidone Iodine 10%		*p* (Kolmogorov-Smirnov)	*p*
**N**		31		37			>0.05 ^x^
**Sex**	**Male**	20		27			
	**Female**	11		10			
**Age**	**Mean ± SD**	38.03 ± 11.12		41.49 ± 11.18		>0.05	>0.05 ^t^
**Area**	**Mean ± SD**	13.13 ± 5.03%		12.11 ± 6.54%		<0.05	>0.05 ^u^
**Inpatient days**	**Mean ± SD**	10.45 ± 6.15		9.92 ± 6.08		<0.05	>0.05 ^u^
**Number of dressing changes**	**Mean ± SD**	1.35 ± 0.66		5.22 ± 3		<0.05	<0.05 ^u^
**Pain during dressing changes**	**Mean ± SD**	Visit 1	1.23 ± 0.76	Visit 1	2.65 ± 0.89	<0.05	<0.05 ^x^
		Visit 2	0.26 ± 0.51	Visit 2	2.24 ± 0.92	<0.05	<0.05 ^x^
		Visit 3	0.03 ± 0.18	Visit 3	1.51 ± 1.10	<0.05	<0.05 ^x^
		Visit 4	0	Visit 4	1.16 ± 1.12	<0.05	<0.05 ^x^
		Visit 5	0	Visit 5	0.7 ± 0.89	<0.05	<0.05 ^x^
**Epithelization**		23		24			>0.05 ^x^
**Absence of epithelialization**		8		13			

^t^ Student’s *t*-test; ^u^ Mann–Whitney U-test; ^x^ Pearson’s chi-squared test.

## Data Availability

Not applicable.

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
