# Peer review of "Therapeutic Treatment of 2A Grade Burns with Decellularized Bovine Peritoneum as a Xenograft: Multicenter Randomized Clinical Trial"

_medicina, 2022, doi:10.3390/medicina58060819_

Round 1

Reviewer 1 Report

Line 55 – Add appropriate marks to product names (i.e., TM, ®, etc.)

Why are there 1 sentence statements that are not part of a paragraph? This occurs throughout the manuscript. Maybe it was formatting issue from creating the PDF?

“Gauze dressings impregnated with Povidone-Iodine” – What product is this and how common is its use on these types of burns? I don’t believe this is typical SOC.

Line 156 – “The patients in both groups underwent mechanical debridemment of the wound from external contaminants and exfoliated epithelium”. Are the wounds debrided to a bleeding wound bed prior to being treated?

Line 159 – What is meant by “surface bandaging with a gauze bandage”? So DBPs were not sutured and/or stapled into the wound bed?

Line 162 – How common was an allergic reaction?

Line 171 – “The presence or absence of complete discharge of the xenograft from the wound was assessed.” Are you referring to graft loss or the product not integrating into the wound?

Line 176 – suggest remove impregnated.

Need to add a conjunction (i.e., and, or, but, etc. in several spots. Lines 125; 165/166; 186/187; 195/196; 334/335

The statement on line 193/194 in unclear. Please revise.

Line 214-216 “In the test group, the change of dressings was considered to be one or more sheets of DBP replaced. The time between dressing changes was not fixed and did not occur in all patients.” Not sure what the 1st sentence is trying to state. There is mention that the control group had dressing changes due to them becoming soaked. Did this occur for the test group? How are you accounting for the difference in time between dressing changes between the groups?

What is the message to be gained from Table 2? It is not a summary of the findings. It appears to be an example. Can Table 1 and 2 be combined somehow?

Check the references to the Tables.

A couple of places have the significance listed as p~0.00. I don’t think this customary. Can it be listed as p<0.05 or even less (i.e., p<0.01,etc.) if appropriate.

Spell check: line 156: debridemment, Figure 2: gause, Table 3: Deccellularized, Povidon, line 315: xenograph

Use decimal for 0.05 instead of comma and don’t need periods after the last number (0.50..)

Please check verb tense agreement throughout. An example on line 319: “Human acellular dermal matrix, in the acute phase of burns, applied for 10 days.” …was applied..; ….has been applied…..; …..has shown successful results after being applied for 10 days.

Why restrict your study to 100% re-ep to be considered the criteria for healing?

Do you have permission to use figure 3? I don’t see a disclaimer.

Line 370  - do you mean covering instead of coating?

Line 374 - you state the use of DBP limited the movement of patients but they reported less pain. Can you comment on this? I would expect limited movement may be avoiding pain.

Author Response

Thank you for reviewing our manuscript, for your comments and suggested corrections. We have revised and corrected our manuscript in accordance with your comments in track changes mode. Also we hade some problems with lines accounting, but I hope this circumstances will not affect the quality of our manuscript. 

The whole list of corrections is in uploaded Microsoft Word file. 

Sincerely,

Baurzhan B. Anapiya

PhD student, NC JSC Karaganda Medical University,

40 Gogol str., Karaganda, Republic of Kazakhstan.

Chief of reconstructive surgery unit

Department of multidisciplinary surgery

National Research Oncology Center, Nur-Sultan

Republic of Kazakhstan

Phone: +77072262728

Email: alaydob@gmail.com

Reviewer 2 Report

The authors present a RCT on a topical dressing in burns.

Some minor and major issue have to be corrected and added before further consideration for publication.

line 50 incomplete thickness should be changed to “ partial thickness”

line 53 A modern method of conservative treatment of II-III A degree burns is the use of biological wound dressings [5]. Please correct the “ A”

Line 54   please specify and distinguish temporary dressing and permanent skin replacement, currently they are mixed up.  The manuscript deals  with 2a so only temporary dressing should be discussed. r

Line 61.63 Please replace pricing  for alloderm by prices for Integra because Integra is widely used while alloderm is nor widely used.  See also comment for line 54

Line 70 please give a ref for preclinical studies or give results

 Line 77 see line 191 please give primary endpoint e.g. time to heal

Line 113 was this study listed by ClinicalTrials.gov ???

 Line 122 how was II a burns determined: by observation or Laser Doppler Imaging

Line 135  please replace cattle and cow by bovine throughout the manuscript

Line 136 137 please give name of product and supplier

In the Material section the description of sample size calculation should be given.

Line 292 the pain scale is not linear therefore calculation of means  is not correct..  

Line 297 add “days”

 Line 309 to 319 As in the introduction temporary dressings and permanent dressings and skin substitutes are again mixed. I would strongly suggest to only mention temporary dressing.  Furthermore the discussion section should discuss  the study design : If epithelialization was assessed at 14  an earlier epithelization time point may be missed .  Therefore, the Bovine material must not be inferior; it is the study design which does not allow for a different (may be better) result. Please change/correct and discuss in the discussion.

Line 352  better than what ? please add to this sentence

 Line 362 if you discuss the pain issue you have to mention that in the gauze PVP group more dressing changes were necessary. This  may account for a higher overall pain. Please discuss this as a short coming of the trail protocol.  The question would be whether the first application reduced pain; which I believe it does because PVP is painful when applied.

Line 365  number of patients cannot be a criterion for failed significance of the primary endpoint in a proper designed clinical study. Therefor please give your sample size calculation. see also critic in Material and Methods.

The Abstract  and introduction  has focused on cost. However this issue is not fully addressed in the discussion. What is the price of gauze and PVP as results are not that bad.  What is an estimated prize of the DBP ( you have mentioned a price in your PRS Global open paper) .

In general I would recommend that the manuscript should follow CONSORT criteria and a flow chart should be added to the manuscript.

Author Response

Thank you for reviewing our manuscript, for your comments and suggested corrections. We have revised and corrected our manuscript in accordance with your comments in track changes mode. Also we hade some problems with lines accounting, but I hope this circumstances will not affect the quality of our manuscript.

Regards,

Baurzhan B. Anapiya

PhD student, NC JSC Karaganda Medical University,

40 Gogol str., Karaganda, Republic of Kazakhstan.

Chief of reconstructive surgery unit

Department of multidisciplinary surgery

National Research Oncology Center, Nur-Sultan

Republic of Kazakhstan

Phone: +77072262728

Email: alaydob@gmail.com
